# Reactive Case Detection Strategy for Malaria Control and Elimination: A 12 Year Systematic Review and Meta-Analysis from 25 Malaria-Endemic Countries

**DOI:** 10.3390/tropicalmed8030180

**Published:** 2023-03-18

**Authors:** Ebenezer Krampah Aidoo, Frank Twum Aboagye, Felix Abekah Botchway, George Osei-Adjei, Michael Appiah, Ruth Duku-Takyi, Samuel Asamoah Sakyi, Linda Amoah, Kingsley Badu, Richard Harry Asmah, Bernard Walter Lawson, Karen Angeliki Krogfelt

**Affiliations:** 1Department of Medical Laboratory Technology, Accra Technical University, Accra GP 561, Ghana; felixbotchway@yahoo.com (F.A.B.); gosei-adjei@atu.edu.gh (G.O.-A.); mappiah@atu.edu.gh (M.A.); rduku-takyi@atu.edu.gh (R.D.-T.); 2Biomedical and Public Health Research Unit, Council for Scientific and Industrial Research-Water Research Institute, Accra AH 38, Ghana; frankaboagye71@gmail.com; 3Department of Molecular Medicine, Kwame Nkrumah University of Science & Technology, University Post Office, Kumasi AK 039, Ghana; samasamoahsakyi@yahoo.co.uk; 4Department of Immunology, Noguchi Memorial Institute for Medical Research, University of Ghana, Accra LG 581, Ghana; lamoah@noguchi.ug.edu.gh; 5Department of Theoretical & Applied Biology, Kwame Nkrumah University of Science & Technology, University Post Office, Kumasi AK 039, Ghana; kingsbadu@gmail.com (K.B.); bwalterlawson@yahoo.com (B.W.L.); 6Department of Biomedical Sciences, School of Basic and Biomedical Science, University of Health & Allied Sciences, Ho PMB 31, Ghana; rasmah@uhas.edu.gh; 7Department of Science and Environment, Unit of Molecular and Medical Biology, The PandemiX Center, Roskilde University, 4000 Roskilde, Denmark; 8Department of Virus and Microbiological Special Diagnostics, Statens Serum Institut, 2300 Copenhagen, Denmark

**Keywords:** reactive case detection, infection reservoir, index cases, passive surveillance, malaria control and elimination

## Abstract

Reactive case detection (RACD) is the screening of household members and neighbors of index cases reported in passive surveillance. This strategy seeks asymptomatic infections and provides treatment to break transmission without testing or treating the entire population. This review discusses and highlights RACD as a recommended strategy for the detection and elimination of asymptomatic malaria as it pertains in different countries. Relevant studies published between January 2010 and September 2022 were identified mainly through PubMed and Google Scholar. Search terms included “malaria and reactive case detection”, “contact tracing”, “focal screening”, “case investigation”, “focal screen and treat”. MedCalc Software was used for data analysis, and the findings from the pooled studies were analyzed using a fixed-effect model. Summary outcomes were then presented using forest plots and tables. Fifty-four (54) studies were systematically reviewed. Of these studies, 7 met the eligibility criteria based on risk of malaria infection in individuals living with an index case < 5 years old, 13 met the eligibility criteria based on risk of malaria infection in an index case household member compared with a neighbor of an index case, and 29 met the eligibility criteria based on risk of malaria infection in individuals living with index cases, and were included in the meta-analysis. Individuals living in index case households with an average risk of 2.576 (2.540–2.612) were more at risk of malaria infection and showed pooled results of high variation heterogeneity chi-square = 235.600, (*p* < 0.0001) I^2^ = 98.88 [97.87–99.89]. The pooled results showed that neighbors of index cases were 0.352 [0.301–0.412] times more likely to have a malaria infection relative to index case household members, and this result was statistically significant (*p* < 0.001). The identification and treatment of infectious reservoirs is critical to successful malaria elimination. Evidence to support the clustering of infections in neighborhoods, which necessitates the inclusion of neighboring households as part of the RACD strategy, was presented in this review.

## 1. Introduction

Malaria, a vector-borne disease, remains a major public health problem. Globally, 84 countries with malaria endemicity recorded an estimated 247 million cases of the disease in 2021, up from 245 million in 2020, with most of this rise occurring in countries in the World Health Organization African Region [1]. However, the malaria elimination effort is gathering momentum in many countries. The total number of malaria-endemic countries that counted fewer than 100 malaria cases increased from 6 in 2000 to 27 in 2021 [1]. During the same period, the number of countries that recorded fewer than 10 indigenous cases rose from 4 to 25 [1].

Over the years, milestones such as these have hinged on vector-control strategies such as periodic indoor residual spraying (IRS), distribution of long-lasting insecticide nets (LLINs), increased funding, strengthening of health systems, seasonal malaria chemoprevention in children, preventive chemotherapies (e.g., intermittent preventive treatment in infants and pregnant women), and improved case reporting and surveillance [2]. Today, the recommendation by the WHO on the use of the RTS, S malaria vaccine for the prevention of *Plasmodium falciparum* malaria in children living in moderate- to high-transmission areas, as described by the WHO [1], will further complement these existing interventions. Some countries focus their malaria prevention strategies on the above-mentioned interventions together with malaria control programs that aim at a decline in the disease burden until it ceases to be a public health problem. Other countries with fewer than 10,000 malaria cases work towards elimination to guarantee sustained zero local transmission of malaria in the population within a specified geographic area through enhanced surveillance systems.

Asymptomatic malaria infections have existed for many years, and finding and treating individual asymptomatic infections, which comprise 60% of the infected population, remains a challenge [3]. An infected asymptomatic individual can become an important reservoir of transmission to healthy individuals under suitable conditions [4] and present an obstacle to the elimination of the disease. In hypoendemic areas moving towards malaria elimination, asymptomatic malaria must be scrutinized within the wider context of sustainable malaria control and elimination strategies. In addition to an increasing use of more sensitive molecular diagnostic methods to achieve elimination and prevent resurgence, surveillance systems will hinge on which strategy is best suited to identify these asymptomatic infections.

Reactive case detection, whereby household members, neighbors, and other contacts of index cases are screened for infection and treated with antimalarial drugs [3], is a strategy recommended by the WHO. Despite the WHO’s recommendation of RACD [5] and its use, knowledge gaps still exist in its implementation. These include questions to do with standard metrics on the coverage of screening needed to affect transmission, optimal target populations, timing, and frequency. Among the Asia Pacific Malaria Elimination Network (APMEN) partner countries, some countries have reported using a specific screening radius (maximum of 2.5 km) around index cases [6]. Procedures to determine neighboring households for investigation have included screening individuals residing within a particular radius of the index case and choosing a definite number of neighbors or households for follow-up [7]. Another study recommended the use of tablets loaded with satellite images from study sites to determine the proportion of households that should be tested [8]. At the individual country level, limited practical know-how exists to inform control programs. RACD can, however, be better executed when operational experiences from different countries are brought together to inform country-specific needs. Hence, the objective of this review is to discuss and highlight RACD as a recommended strategy for the detection and elimination of asymptomatic malaria as it pertains in different countries. Few RACD reviews and meta-analyses that include research published up to the year 2022 currently exist. Newby et al. [9] and Perera et al. [10] reviewed the literature up to 2018 and 2019, respectively. Subsequently, a review by Stresman et al. [11] and a meta-analysis by Deen et al. [12] (confined to the Greater Mekong Sub-region) up to 2019 and 2021, respectively, were undertaken. A review and meta-analysis will expand the frontiers of current knowledge for programs geared towards malaria elimination by this strategy.

## 2. Materials and Methods

### 2.1. Data Sources and Search Strategy

A systematic review and meta-analysis of RACD strategies for malaria control and elimination was conducted per the Preferred Reporting Items for Systematic Reviews and Meta-Analyses (PRISMA) guidelines [13]. Search terms (combined free text and keywords) included “malaria and reactive case detection”, “contact tracing”, “focal screening”, “case investigation” and “focal screen and treat”. Relevant studies published between January 2010 and September 2022 were identified mainly through PubMed and Google Scholar. The retrieved studies were manually screened to identify the relevant studies.

### 2.2. Study Selection

Eligibility for inclusion was determined according to the following selection criteria: reported results of the RACD strategy (including number of cases followed, number of persons traced, number of new cases detected). The following information (Table 1) was extracted: (1) year of publication; (2) country of study; and (3) time period. Studies restricted to simulation, modeling, resampling algorithms, and articles that did not discuss RACD specifically were excluded. Review articles were also excluded from the analysis. Fifty-four (54) studies were systematically reviewed. Of these studies, 7 met the eligibility criteria based on risk of malaria infection in individuals living with an index case < 5 years old, 13 met the eligibility criteria based on risk of malaria infection in an index case household member compared with a neighbor of an index case, and 29 met the eligibility criteria based on risk of malaria infection in individuals living with index cases, and were included in the meta-analysis. A summary of the RACD strategies and outcomes, including the timing of the RACD, the baseline study participant characteristics, the RACD households, and the screening radius are shown in Table 1 and Appendix A.

### 2.3. Data Extraction and Assessment of Study Quality

Duplicates were removed with EndNote Reference Library. Studies that met the eligibility criteria were included after going through careful screening and evaluation by two reviewers (EKA and FTA). Studies were initially considered depending on their titles and abstracts, and this was followed by a thorough review of the complete article for relevance. To clarify any disagreements, a third investigator (SAS) was involved.

### 2.4. Statistical Analysis

Statistical analysis was carried out using MedCalc 20.0 (MedCalc Software Ltd., Ostend, Belgium). The findings from the pooled studies were reported as odds ratios with their confidence intervals from a pooled fixed-effect model. A Forest plot was generated from the outcomes to graphically represent the results. Publication bias and heterogeneity were tested using Egger’s test and Higgins I^2^ [67], respectively. For I^2^, values between 25% and 50% indicated low heterogeneity, values between 50% and 75% indicated moderate heterogeneity, and values greater than 75% indicated severe heterogeneity. A p-value of ≤ 0.05 was considered significant in all cases.

## 3. Results

### 3.1. Search Result

In all, 115 studies were identified through PubMed and 284 studies were identified though Google Scholar (Figure 1). Following the removal of duplicates, 260 studies were screened, and this left 154 studies for full assessment. Of these, 100 studies were excluded because they concerned strategies other RACD (e.g., simulation, modeling). The remaining studies were carefully evaluated in detail, resulting in a total of 54 studies for qualitative synthesis and 7, 13, and 29 studies for quantitative synthesis.

### 3.2. Study Characteristics and Quality Assessment

All 7, 13, and 29 respective studies used for the meta-analysis involved real RACD strategies. In all, the studies included a total of 8965 index cases leading to 90,940 contacts (Appendix A). While 16 studies used only RDT to test contacts, 2 studies used RDT/PCR, 1 study used LAMP/PCR, 1 study used microscopy/PCR, 6 studies used PCR, 1 study used microscopy, 2 used studies LAMP, and 5 studies did not report the diagnostic method used (Appendix A). Index case demographics and individual study characteristics are shown in Appendix A. Although some of the included studies had a higher risk of bias than others, sensitivity analyses that excluded those studies showed significant difference. Quality assessment was independently carried out by two reviewers, and any disagreements were resolved by discussion.

### 3.3. Outcomes

#### 3.3.1. Risk of Malaria Infection Amongst Persons Living with Index Cases

Twenty-Nine studies met the eligibility criteria for the meta-analysis (Figure 2). The average risk of malaria infection for an individual living with an index case was 2.576 (2.540–2.612) and was statistically significant (*p* = 0.033). The risk of infection for an individual living with an index case ranged between 0.032 and 232.545. From the pooled results, high variation heterogeneity chi-square = 235.600, (*p* < 0.0001) I^2^ = 98.88 [97.87–99.89] (assessed with Egger’s Test, intercept = 1.154, *p* = 0.8534; test for overall effect, Z = 2.136, (*p* = 0.003) was observed. 

#### 3.3.2. The Risk of Malaria Infection in a Neighbor of an Index Case Compared with an Index Case Household Member

The pooled results from 13 studies indicated that neighbors of index cases were 0.352 [0.301–0.412] times more likely to have a malaria infection relative to index case household members, and this was statistically significant (*p* < 0.001) (Figure 3). The risk of infection from the pooled fixed effect model ranged between 0.029 and 4.062, and, from the results, high variation heterogeneity chi-square = 285.247, (*p* < 0.0001) I^2^ = 95.79 [94.81–96.96] (assessed with Egger’s Test, intercept = 2.989, *p* = 0.903; test for overall effect, Z = 2.533, (*p* = 0.011)was observed.

#### 3.3.3. Risk of Malaria Infection Living with an Index Case < 5 Years Old

The risk of malaria infection in an individual living with an index case < 5 years old according to the pooled fixed effect model ranged between 0.240 and 18.520. The average risk of infection was 3.882 (1.526–4.617) according to the pool of seven studies that were statistically significant (*p* = 0.003) (Figure 4). According to the pooled results, high variation heterogeneity chi-square = 25.47, (*p* < 0.0001) I^2^ = 93.89 [91.36–96.96].

### 3.4. Publication Bias

Publication bias was not assessed for the outcome in Figure 4 involving the 7 studies for which a funnel plot was not retrieved [68], but the other 2 outcomes had their publication bias assessed [69].

### 3.5. Sensitivity Analysis

A sensitivity analysis was performed for the risk of malaria infection in a neighbor of an index case compared with index case household members to deal with heterogeneity. After excluding five studies (Conner et al. [26], Stuck et al. [33], Aidoo et al. [37], Hustedt et al. [53], and Fontoura et al. [54]), the analysis was performed again to generate a forest plot (Figure 5). A change from 0.352 [0.301–0.412] to 0.459 [0.294–0.715] in the final odds ratio was observed, and heterogeneity was reduced from 95% to 46% when these studies were excluded.

## 4. Discussion

Malaria elimination calls for strategies aimed at reducing parasite reservoirs, both in hypoendemic and hyperendemic areas. RACD strategies are utilized primarily by malaria elimination programs and have been implemented in Asia, South America, and Africa (with mixed results). RACD application in locations in Africa or elsewhere may differ by way of coverage as a result of differences in human behavior, settlement patterns, and epidemiology of malaria transmission. RACD may provide a strategy to detect malaria hotspots. However, challenges may exist due to low parasite reservoir coverage despite large operational efforts.

The RACD studies under review examined factors relevant to malaria elimination, including diagnosis, infection reservoir (asymptomatic, submicroscopic), prevalence, surveillance, and elimination strategy. Data pooled from the different RACD studies were utilized to calculating the average risk of malaria infection among household members and neighbors. The pooling of quantitative data from regions with varying malaria endemicity may not truly reflect the risk of infection for a particular region as this will be affected by factors such as vector abundance, socioeconomic circumstances, malaria health policies in the regions, etc., and thus may differ accordingly.

In endemic regions, malaria transmission is often seasonal [70] and heterogeneous [71]. To determine whether the spatial targeting of high-risk areas is of concern, malaria control and elimination programs must understand the extent of the community-, district-, and regional-level spatial heterogeneity of malaria risk. By identifying hotspot locations, it is possible to quantify spatial risk and identify the particular risk factors that may be causing an increase in infection risk in those places. Malaria distribution in endemic countries is highly heterogeneous. From our meta-analysis, the distinct epidemiological features of malaria in different countries could be related to the significant heterogeneity in different transmission settings. Since RACD is best suited for hypoendemic areas, it could be possible that when a local area gets closer to eliminating local malaria, increased clustering is anticipated, and that these clusters may show particular features that are challenging to control. A RACD strategy that entails distinct transmission processes for local elimination can be utilized to further reduce malaria clusters in hypoendemic areas. However, in countries with a high malaria burden, routine strategies for case reduction can be continued or scaled up [72]. Per the results of the meta-analysis, the risk of malaria infection in individuals living with an index case, malaria infection risk in neighbors of an index case compared with index case household members, and malaria risk in individuals living with an index case < 5 years old were significant.

Using spatial extent or screening radius as a RACD implementation metric, numerous studies have indicated a variety of strategies [32,37,50,61]. While some investigations only involved household members [46], others expanded their investigation to include index case neighbors within varying distances [50,53]. Malaria transmission has been reduced as a result of interventions aimed at both entire communities and households with known malaria infections [30,73]. The majority of the positive cases identified by RACD were geographically found near index case households. Many African cities show a clear trend of increasing malaria transmission from urban to peri-urban to rural settings [74]. The spatial extent of RACD in urban Lusaka, where population density is higher and more people are tested, was associated with a higher probability of identifying additional cases within 250 m of uninhabited land [25]. Urban malaria transmission may vary depending on an assortment of factors, such as vector breeding sites, local vector species, location, human movement patterns, waste management, socioeconomic factors, land use, climate, and local malaria intervention programs [75]. In cases of low transmission, strategies that target the household are thought to be operationally sustainable. A RACD strategy targeting households offers a practical programming alternative given that achieving elimination would require sustained efforts over the long run.

Secondary infections are detected where neighbors have been involved in a RACD strategy, and some studies have shown significant declines in positivity as distances increase [8,32]. However, the operational practicality is constrained by the low numbers of secondary infections detected in field investigations, which necessitates screening a large number of people [76]. Here, we provided evidence to back up the notion of infection clustering in neighborhoods, which in turn makes the inclusion of neighboring households a part of the RACD strategy. Nevertheless, the ideal screening radius from the index case household to the neighboring households as part of RACD still remains unclear. The determination of a screening radius is informed by the local malaria epidemiology (including vector dispersal) and logistical capacity [5]. Targeting high household coverage over a smaller screening radius in unsprayed areas would enhance efficiency and responsiveness and help establish an ideal screening radius. While in Swaziland, a 1 km screening radius was operationally demanding, in some APMEN partner countries a maximum screening radius of 2.5 km was covered.

According to Sturrock et al. [8], RACD implementation metrics, such as response time, indicated that the odds of detecting secondary infections were much higher if RACD was triggered within a week of presentation of the index case. Such rapid response ensures that asymptomatic cases are identified prior to becoming symptomatic and seeking treatment. This allows only a limited opportunity to infect mosquitoes. While performing RACD, Aidoo et al. [37] screened 1280 individuals living in 413 households., and in the 100,000 individuals in the catchment area, an estimated total of 12,900 infections were present at any time. RACD identified a total of 144 additional cases in index case and neighbor households. However, the screening of 1000–2400 individuals would have led to the identification of 200–500 secondary cases each month. This translated into 1.5–3.9% of all asymptomatic infections relative to the estimated total of 12,900 infections in the catchment population of the hospital. To estimate the overall number of infections identified during an extended period, such as a whole year, it would be necessary to have a thorough understanding of the duration of infections, the total number of people infected over time, and the temporal variations in transmission foci. The odds of finding secondary cases were lower when the index household was sprayed. Where data on neighboring households are not available, spraying is normally directed at predefined locations rather than individual houses. Since index households could serve as indicators for spraying locations, its protective effect could possibly be extended to neighboring households, leading to a reduction in the local vector density. In addition, it is possible that infections in the sprayed regions are acquired while engaging in recreational or work-related activities outside the house. RACD should only be initiated in receptive areas where there is a likelihood of transmission around the vicinity of the index case. In these locations, RACD should be employed irrespective of whether a case has been imported or locally transmitted because imported cases can result in local transmission [3]. While some RACD studies concentrated only on symptomatic individuals, others focused on both symptomatic and asymptomatic individuals. In all these RACD studies, many more asymptomatic infections (mostly sub-microscopic) were identified compared with symptomatic infections. In hypoendemic areas, asymptomatic infections are common, the majority of which are sub-microscopic [77] and constitute up to 50% of human-to-mosquito transmissions [78]. Nevertheless, it is possible that hotspots of mainly asymptomatic infections were missed because symptomatic infections (index cases which triggered RACD) and asymptomatic infections do not necessarily overlap spatially [79]. Asymptomatic carriers can act as reservoirs of infection due to their failure to seek treatment [80], further derailing elimination efforts. The challenges of asymptomatic infections are also compounded by diagnostic limitations [81]. RACD strategies employing malaria diagnostic tools of high sensitivity supplement the everyday passive case detection in hypoendemic settings and may be used in malaria elimination programs. Thus, alternative diagnostic tools that are more field-friendly and sensitive, such as highly sensitive RDTs (HS-RDTs) or loop-mediated isothermal amplification (LAMP), need to be assessed and used for the identification of sub-microscopic/asymptomatic malaria. In lieu of this, presumptive treatment targeting vulnerable individuals during the high transmission seasons may be a strategy that can help address the challenge of low parasitaemia missed by RDTs and poor RACD screening coverage [82].

The implementation of RACD differed with the different subgroups tested (e.g., index case household only, index case household versus neighbors, index case household versus neighbors plus control group), the number of index cases that triggered RACD, and the radius covered. Though most studies reported adhering to case investigation data collection guidelines, there was a collection of variables that each study decided to analyze, and this reflected the tailoring of investigations to local conditions and capacies. A fully operational surveillance and response system is required for malaria elimination, in addition to at least three consecutive years with no indigenous malaria cases [83]. It has long been acknowledged that the elimination of malaria requires effective malaria surveillance [83]. With a workforce committed to surveillance at all levels of the health system [84] and integrated response mechanisms [85], countries that have successfully eliminated malaria have typically used a combination of effective passive case detection (PCD) [86] and active case detection (ACD) activities [84]. According to China’s 1-3-7 framework, malaria cases must be reported within one day, cases must be investigated within three days, and foci investigation and increased preventive measures must be conducted by day seven [87]. Depending on the risk and endemicity levels, different responses are recommended, including “intensified surveillance and response” in border areas, “passive surveillance during the transmission season and active surveillance targeting transmission foci” in zones with seasonal malaria, and “active and passive surveillance, with special attention to mobile populations,” in areas with higher incidence [88]. The Immediate Disease Notification System (IDNS), a surveillance system with notifiable disease system and surveillance outputs that swiftly transmit to a team to trigger a response, is part of the malaria surveillance system in Swaziland [89]. It would be ideal for countries that have eliminated malaria to share their individual experiences. In endemic countries, sharing data on population movements across borders will make it easier to eliminate the disease. This strategy, in our opinion, would be beneficial for regional cooperation in the fight against malaria [89].

Operational gaps exist concerning which RACD approach is most effective at reducing and preventing transmission. This review recommends the use of standardized data metrics in assessing the RACD screening radii, the number of individuals tested, the completeness of geographical coverage during RACD, and the timeliness of the incorporated activities. The integration of more reliable data and the development of maps of foci of transmission by geo-locating cases to the households can support the targeting of these vulnerable individuals and their neighborhoods with malaria interventions [90,91].

## 5. Conclusions

Despite the WHO surveillance benchmarks for elimination in endemic regions and the operational know-how of affected countries, it is not likely that recommendations for one setting will apply to all others across diverse epidemiological areas. The choice of whether or not to use a RACD strategy is dependent on an assortment of factors including local transmission intensity, cost, operational feasibility, and population receptiveness. As countries progress towards elimination, malaria programs should prioritize case investigation and undertake RACD to identify remaining reservoirs of infection.

## Figures and Tables

**Figure 1 tropicalmed-08-00180-f001:**
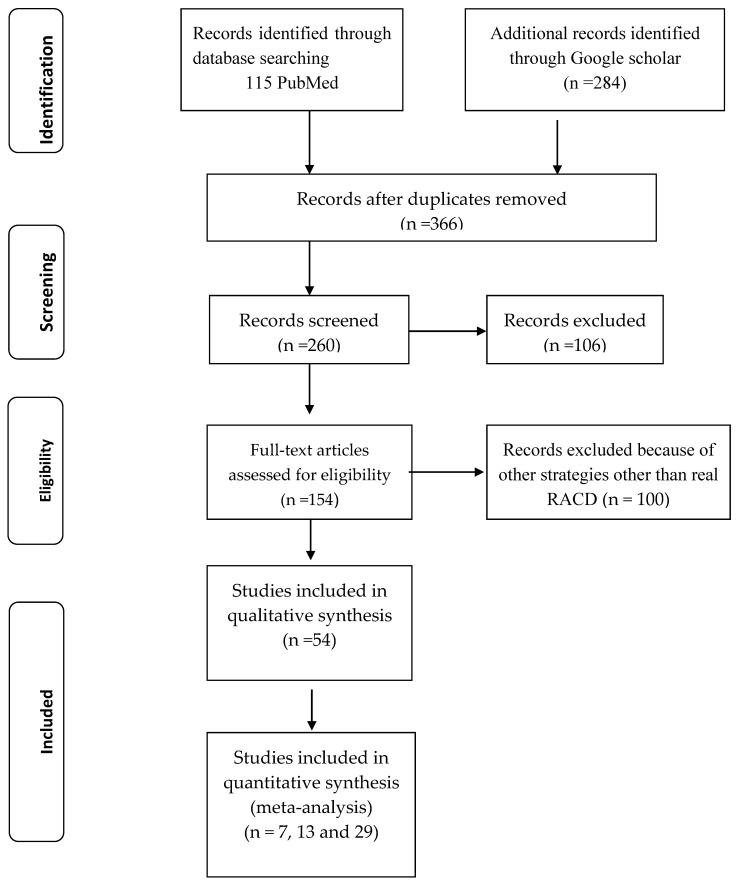
PRISMA flow diagram showing schematic illustration of database searches, identification, screening, and eligibility of included studies.

**Figure 2 tropicalmed-08-00180-f002:**
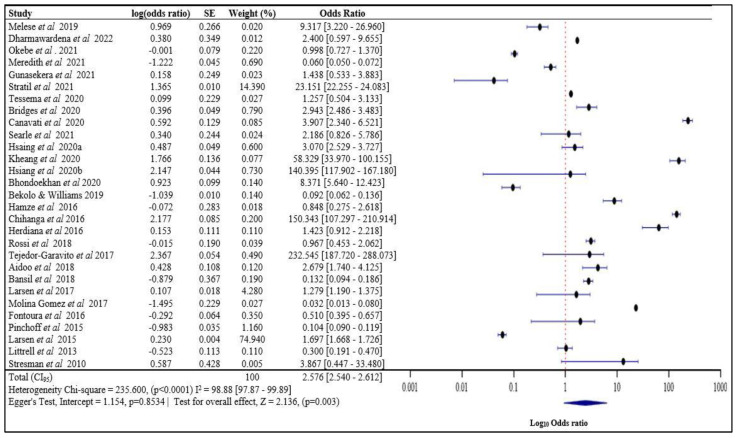
Risk of malaria infection amongst persons living with index cases [14,15,17,18,19,23,24,25,29,30,31,32,34,35,36,37,41,46,47,50,51,52,54,55,57,62,63,64,66]. SE: Standard Error, Log(odds ratio) values with “-“ are negative values.

**Figure 3 tropicalmed-08-00180-f003:**
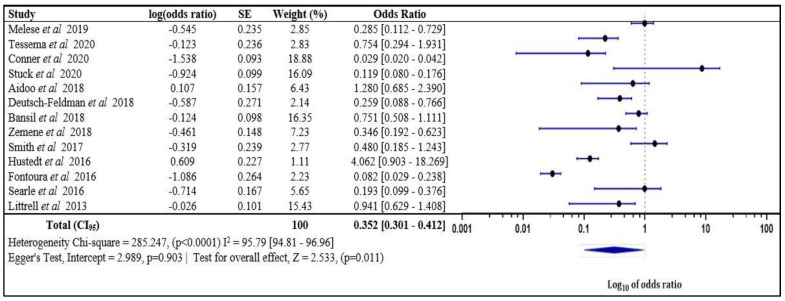
The risk of malaria infection in a neighbor of an index case compared with an index case household member [24,26,33,36,37,38,41,43,49,53,54,58,64]. SE: Standard Error, Log(odds ratio) values with “-“ are negative values.

**Figure 4 tropicalmed-08-00180-f004:**
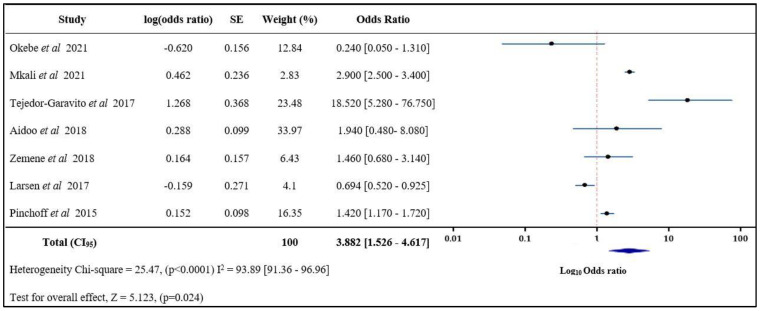
Risk of malaria infection in individuals living with an index case < 5 years old [15,20,37,43,47,51,62]. SE: Standard Error, Log(odds ratio) values with “-“ are negative values.

**Figure 5 tropicalmed-08-00180-f005:**
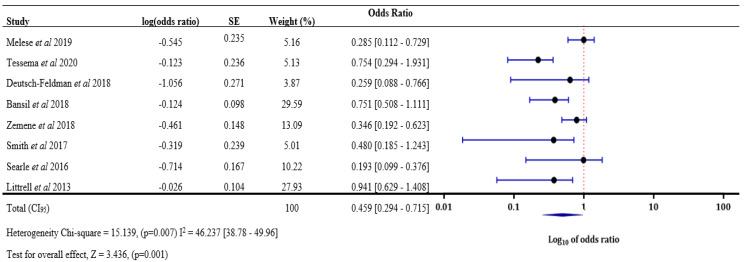
Malaria infection risk in neighbours of index cases compared with index case household members after sensitivity analysis [24,36,38,41,43,49,58,64]. SE: Standard Error, Log(odds ratio) values with “-“ are negative values.

**Table 1 tropicalmed-08-00180-t001:** Summary of included studies.

Study, Country	Time/Period	Malaria Transmission	Main spp.	Source of Index Cases	Trigger and Malaria Test	Spatial Extent	RACD Response Per Protocol	RACD Response in Real Time (Reality)
Dharmawardena et al., 2022 [14], Sri Lanka	2015–2016 (months not stated)	Low transmission	Pf and Pv	Cases imported by AMC	Index case in health facility (test not reported)	Not reported	Not reported	Not reported
Okebe et al., 2021 [15], Gambia	June 2016–December 2018	Seasonal transmission	Pf	Index case reported by village health workers/community	Index case reported by village health workers (test not reported)	Index case household	Not reported	Within 5 days of index case being reported
Roh et al., 2021 [16], Thailand	November 2017–September 2018	Seasonal transmission	Pf and Pv	1 health facility	Index case in health facility (RDT)	50 other at-risk persons associated with the index case	Within 7 days of index case being reported	Mainly within 7 days (6% involved within 10 days)
Meredith et al., 2021 [17], Kenya	August 2018–October 2019	Low transmission	Pf	6 health facilities	Index case in health facilities (RDT)	Index case household	Not reported	Not reported
Gunasekera et al., 2021 [18], Sri Lanka	2017–2019 (months not stated)	No indigenous transmission	Not reported	Public and private health facilities	Index case in health facility (microscopy)	Index case household and neighboring households within 1 km radius	Within 2 days in vicinity of index case (primary RACD); 3–4 weeks in neighboring households (secondary RACD)	Within 2 days in vicinity of index case (primary RACD); 3–4 weeks in neighboring households (secondary RACD)
Stratil, et al., 2021 [19], Cambodia	January 2018–December 2020	Low transmission	Pf	Mobile malaria posts	Index case in mobile malaria post reported by mobile malaria workers (test not reported)	Co-travellers of confirmed index case	Not reported	Not reported
Mkali et al., 2021 [20], Tanzania	August 2012–December 2019	Seasonal transmission	Not reported	189 public and 124 private health facilities	Index case in health facility (RDT and microscopy)	Index case household	Not reported	Not reported
Vilakati et al., 2021 [21], Eswatini (Swaziland)	September 2015–August 2017	Low transmission	Pf	287 public or private health facilities	Index case in health facility (RDT and microscopy)	households within 500 m of index case household	Not reported	Within 7 days (at most 5 weeks) of index case presentation
Morales et al., 2021 [22], Ecuador	April 2019–February 2020	Low transmission	Pv	Health facilities	Index case in health facility (microscopy)	3 km around households of index cases involving 6 neighborhoods	Not reported	Not reported
Searle, et al., 2021 [23], Zambia	March 2016– July 2018	Low transmission	Pf	Community health worker	Index case reported by community health worker (RDT)	Index case household and neighboring households within 250 m	Not reported	A day after the notification visit to the index case household
Tessema et al., 2020 [24], Ethiopia	October 2016–December 2016	Low transmission	Pf and Pv	2 health posts	Index case in health post (RDT)	Index case household, 6 nearest neighbors, and controls	Not reported	Within 2 days of the index case being reported
Bridges et al., 2020 [25], Zambia	2011–2015 (months not stated)	Seasonal transmission	Pf	27 health facilities	Index case in health facility (RDT or microscopy)	Index case household and 9 closest neighboring households	Within 1 week of the index case being reported	RACD is triggered if the index case does not report a travel history within the previous month
Conner et al., 2020 [26], Senegal	October 2014–December 2014	Low transmission	Pf	Health posts/community health workers	Index case in health post and reported by community health workers (test not reported)	Index case household and 5 closest neighboring households within 100 m radius	Not reported	The mean number of days between index case detection and the start of the household visits was 1.3
Grossenbacher et al., 2020 [27], Tanzania	June 2017–August 2018	Low transmission	Pf	16 public and 8 private health facilities	Index case in health facility (RDT or microscopy)	Index case household, 4 nearest neighbors, and 5 households within 200 m of the index case household	Within 1 day for case classification and within 3 days for treatment of infected household members	Within 3 days of index case being reported
Daniels et al., 2020 [28], Senegal	September 2012–December 2015	Low transmission	Pf	Health facility	Index case in health facility (RDT)	Index case household and 5 closest neighboring households	Not reported	Within 3 days of index case being reported
Canavati et al., 2020 [29], Vietnam	September 2016–October 2016	Seasonal transmission	Pf and Pv (tested for but none found)	Health centers	Index case in health center (RDT or microscopy)	Index case and neighbors from forest and farm huts within 500 m of the index cases.	Not reported	Not reported
Hsiang et al., 2020 [30], Namibia	January 2017–December 2017	Low transmission	Pf	11 health facilities	Index case in health facility (RDT or microscopy)	Index case household and neighboring households within 500 m	Within 7 days to 5 weeks of the index case being reported	Within 5 weeks of the index case being reported
Kheang et al., 2020 [31], Cambodia.	July 2015–January 2017	Low transmission	Pf and Pv	Village malaria workers or health facilities	Index case in health facility / reported by village malaria workers (RDT or microscopy)	Testing of co-travellers, index case household, and neighboring households with suspected malaria cases	Within 7 days of the index case being reported	Within 3 days of the index case being reported
Hsiang et al., 2020 [32], Eswatini (Swaziland)	September 2012–March 2015	Low transmission	Pf	261 public or private health facilities	Index case in health facility (RDT or microscopy)	Index case household and neighboring households within 500 m	Within 5 weeks of the index case being reported	Within 2 days of the index case being reported
Stuck et al., 2020 [33], Tanzania	May 2017–January 2018 and June 2018–October 2018	High and Low transmissions	Pf	154 public health facilities or 50 private facilities	Index case in health facility (RDT)	Index case household, 4 nearest neighbors, and 5 households within 200 m of the index case household	Not reported	within 3 days of the index case being reported
Bhondoekhan et al., 2020 [34], Zambia	January 2015–July 2017	Low transmission	Pf	13 health centers and 23 health posts	Index case in health facility (RDT)	Index case household and neighboring households within 250 m	Not reported	within 1 week of the index case being reported
Bekolo and Williams, 2019 [35], Cameroon	April 2018–June 2018	High transmission	Pf	12 primary and nursery schools, 4 health centers, and 13 community neighborhoods	Index case in primary school, nursery school, health center, and community neighborhoods (RDT)	Members of index case household who had fever in the past week	Not reported	Within 1 week of the index case being reported
Melese et al., 2019 [36], Ethiopia	February 2019–April 2019	Seasonal transmission	Pf and Pv	2 health centers	Index case in health facility (test not reported)	Index case household and neighboring households within 200 m	Not reported	Not reported
Aidoo et al., 2018 [37], Kenya	October 2015– August 2016	Low transmission	Pf	1 health facility	Index case in health facility (microscopy)	Index case compound, 5 neighboring compounds, and 5 control compounds	Not reported	Within 7 days of the index case being reported
Deutsch Feldman et al., 2018 [38], Zambia	January 2015– March 2016	Low transmission	Pf	Health center	Index case in hospital (RDT)	Index case household and neighboring households within 250 m	Not reported	Within 7 days of the index case being reported
Kyaw et al., 2018 [39], Myanmar	January 2016– December 2016	Low transmission	Pv, Pf, and Po	Health facility and community level	Index case reported by village health volunteers/basic health staff (RDT)	Not reported	Within 7 days of the index case being reported	Within 7 days of the index case being reported in 95.5% of individuals
Zelman et al., 2018 [40], Indonesia	May 2014–December 2015	Low transmission	Pv, Pf, and Pk	Health facilities	Index case in health facility (microscopy)	Index case household and neighboring households within 500 m	Not reported	Within 7 days of the index case being reported
Bansil et al., 2018 [41], Ethiopia	October 2014–February 2015	Low transmission	Pf and others	213 health centers	Index case in health center (RDT)	Index case household and 10 neighboring households within 100 m radius	Not reported	Within 30 days of the index case being reported
Feng et al., 2018 [42], China	2015	Low transmission	Pv and Pm	Community	Locally acquired case (test not reported)	Index case household and neighboring households	Not reported	Within 7 days of the index case being reported
Zemene et al., 2018 [43], Ethiopia	June 2016–November 2016	Low and seasonal	Pf and Pv	2 health centers	Index case in health center (microscopy)	Index case household and neighboring households within 200 m radius	Not reported	Within 7 days of the index case being reported (typically within 3 days)
Naeem et al., 2018 [44], Pakistan	January 2015–December 2015	Low transmission	Pf and Pv	Military hospital	Index case in hospital (microscopy)	Index case and control households in the vicinity with similar socio-economic status	Not reported	Not reported
Zhang et al., 2018 [45], China	2013–2016 (months not stated)	Low transmission	Pf and Pv	Health facilities	Index case in health facility (RDT/Microscopy/PCR)	Index case contacts, such as coworkers who travelled to the same area (inactive foci), family members, neighbors, and others (active foci)	Not reported	Not reported
Rossi et al., 2018 [46], Cambodia	October 2015–May 2017	Low transmission	Pf	28 pairs of village malaria workers	Index case reported by village malaria worker (RDT)	Index case household	Not reported	Not reported
Larsen et al., 2017 [47], Zambia	2012–2013	Low transmission	Pf	Community health workers and clinics	Community health worker / clinics (RDT)	Index case household and neighboring households within 140 km	Within 7 days of the index case being reported	Not reported
Wang et al., 2017 [48], China	January 2012–December 2014	Low and high transmission	Not reported	12 hospitals	Index case in hospital (test not reported)	Index case household	Not reported	Within 7 days of the index case being reported
Smith et al., 2017 [49], Namibia	January 2013–August 2014	Low and seasonal	Pf	Health facilities in Namibia	Index case in health facility (RDT)	Index case compound, 4 neighboring compounds, and selected controls	Within 2 days of the index case being reported	Within 2 weeks to 2 months of the index case being reported
Molina Gómez et al., 2017 [50], Colombia	Not reported	Seasonal transmission	Pf and Pv	1 hospital	Index case in hospital (test not reported)	Index case household and 4 neighboring households	Not reported	Not reported
Tejedor-Garavito et al., 2017 [51], Swaziland	January 2010–June 2014	Low transmission	Pf	Health facilities	Index case in health facility (RDT)	Index case household and neighboring households within 1 km	Within 5 weeks of the index case being reported	Within 7 days of the index case being reported
Hamze et al., 2016 [52], Democratic Republic of Congo	November 2013–January 2014	High transmission	Pf	1 clinic	Index case in clinic (RDT)	Index case household	Not reported	Not reported
Hustedt et al., 2016 [53], Cambodia	May 2013–March 2014	Low transmission	Pf and Pv	Health facility and community	Index case in health facility and community reported by village malaria workers (RDT)	Index case compound, 5–10 neighboring compounds, and 5 control compounds	Not reported	Within 3 days of the index case being reported
Fontoura et al., 2016 [54], Brazil	January 2013–July 2013	Low transmission	Pv	1 clinic	Index case in clinic (microscopy/qPCR)	Index case compound, 5 neighboring compounds, and 5 control compounds	Not reported	Within 6 months of the index case being reported
Chihanga et al., 2016 [55], Botswana	October 2012–December 2014	Seasonal transmission	Pf	Health facilities	Index case in health facility (RDT or microscopy)	Index case household and neighboring households within 100 m	Within 2 days of the index case being reported	Not reported
Donald et al., 2016 [56], Vanuatu	2013–2014 (months not stated)	Low transmission	Pf and Pv	1 hospital, 11 dispensaries, 4 health centers, and 28 aid posts	Index case in health facility (RDT)	Index case household and neighboring households within 500 m	Not reported	Within 5 days of the index case being reported
Herdiana et al., 2016 [57], Indonesia	June 2014–December 2015	Low transmission	Pv, Pf, and Pk	5 sub-district level primary health centers (PHCs)	Index case in primary health center (microscopy)	Index case household and neighboring households within 500 m	Not reported	Within 7 days of the index case being reported
Searle et al., 2016 [58], Zambia	January 2014–June 2014	Low transmission	Pf	20 rural health centers	Index case reported by community health worker/rural health post (RDT)	Index case household and neighboring households within 140 km	Within 7 days of the index case being reported	Not reported
van Eijk et al., 2016 [59], Chennai (India)	January 2014– January 2015	Low and seasonal	Pf and Pv	1 urban clinic	Index case in clinic (microscopy)	Index case household, contacts in same apartment block, and households within 0.2 km	Index case household screened within 1–7 days, contacts in same apartment block and other households screened within 14 days	Index case household 91.6% screened within 1 week, contacts in same apartment block and other households 64.8% screened within 2 weeks
van Eijk et al., 2016 [59], Nadiad (India)	March 2014–September 2014	Low and seasonal	Pf and Pv	1 urban clinic	Index case in clinic (microscopy)	Index case household, contacts in same apartment block (100 m), and households within 100–1000 m	Index case household screened within 1–7 days, contacts in same apartment block and other households screened within 14 days	Index case household 84.4% screened within 1 week, contacts in same apartment block and other households 93.9% screened within 2 weeks
Wangdi, et al., 2016 [60], Bhutan	2014–2015 (months not stated)	High transmission	Pf and Pv	Health centers	Index case in health center (test not reported)	Index case household and neighboring households within 1 km	Not reported	Not reported
Larson et al., 2016 [61], Zambia	2014 (months not stated)	Not reported	Not reported	173 health facilities	Index case in health facility (RDT)	Index case household and neighboring households	Not reported	Not reported
Pinchoff et al., 2015 [62], Zambia	June 2012–June 2013	Seasonal transmission	Pf	1 clinic	Index case (RDT and microscopy) in clinic	Index case household	Not reported	Not reported
Larsen et al., 2015 [63], Zambia	2014–2015	Low transmission	Pf	Community health workers and 20 clinics	Index case in clinic (RDT)	Index case household and 10 neighboring households	Within 7 days of the index case being reported	Not reported
Littrell et al., 2013 [64] Senegal	2012	Seasonal transmission	Pf	13 clinics	Index case in clinic (test not reported)	Index case compound and 5 neighboring compounds	Within 3 days of the index case being reported	Not reported
Sturrock et al., 2013 [8], Swaziland	December 2009–June 2012	Seasonal transmission	Pf	Health facilities in Swaziland	Locally acquired case or imported case (test not reported)	Index case household and neighboring households within 1 km	Not reported	48.6% screened within 1 week, 87.3% screened Within 14 days.
Rogawski et al., 2012 [65], Thailand	July 2011	Low and seasonal	Pf and Pv	1 hospital	1 case in hospital (test not reported)	Neighbors within 1 km of index case (fever was not a criterion)	Not reported	After 2 weeks of the index case being reported
Stresman et al., 2010 [66], Zambia	June 2009– August 2009	Seasonal transmission	Pf	4 rural clinics	Case (RDT) in clinic	Homestead of index case (fever was not a criterion)	Within 2 weeks of the index case being reported	Not reported

Pm: *Plasmodium malariae*; Po: *Plasmodium ovale*; Pv: *Plasmodium vivax*; Pk: *Plasmodium knowlesi*; Pf: *Plasmodium falciparum*; AMC: Anti Malaria Campaign; RDT: Rapid Diagnostic Test; PCR: Polymerase Chain Reaction; qPCR: Quantitative Polymerase Chain Reaction.

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
