# Peer review of "Reactive Case Detection Strategy for Malaria Control and Elimination: A 12 Year Systematic Review and Meta-Analysis from 25 Malaria-Endemic Countries"

_tropicalmed, 2023, doi:10.3390/tropicalmed8030180_

Round 1

Reviewer 1 Report

Accept in present form

Author Response

Reviewer 1

Accept in present form.

Response: Thanks very much for your approval of our work.

Reviewer 2 Report

Reactive Case Detection Strategy for Malaria Control and Elimination: A 12-Year Systematic Review and Meta-Analysis from 25 Malaria Endemic Countries

Review

In this review, a strategy is demonstrated about evidence to support the clustering of infection in the neighbourhood, which necessitates the inclusion of neighbouring households as part of the RACD. The choice of RACD strategy depends on a gamut of factors, including local transmission intensity, cost, operational feasibility and population receptiveness. As countries progress towards elimination, malaria programmes should prioritize case investigation and undertake RACD to identify remaining reservoirs of infection.

It is a well-written paper with helpful information.

English language quality must be improved.

References

pl. edit all of your refs and correct them. There are a lot of irregularities related to the citation style.

Author Response

English language quality must be improved.

  1. edit all of your refs and correct them. There are a lot of irregularities related to the citation style.

"It is a well-written paper and I do not have additional specific comments. However, the part of references should be edited by the authors and corrected. This is the reason for my decision. Otherwise, the manuscript is ok."

Response: Thanks very much as always. Following your initial comments and subsequent follow up with excerpts of your response indicated above, we have worked on the references to suit the referencing style of the journal

Reviewer 3 Report

Dear authors

Your systematic and meta-analysis review  based on RACD strategies is informative and useful, but as you know detecting the sensitivity and specificity of cases is important to analyze the asymptomatic malaria infections. If possible insert the  items in table 1.     

Author Response

Your systematic and meta-analysis review based on RACD strategies is informative and useful, but as you know detecting the sensitivity and specificity of cases is important to analyze the asymptomatic malaria infections. If possible, insert the items in table 1. 

Response: Thanks very much for your comments. Unfortunately, we are unable to insert the sensitivity and specificity of the cases in table1 because the contents of table 1 were extracted from published studies which made no provision for such information.

Reviewer 4 Report

Reviewer’s comments

The review is focused on Reactive case detection as means of detecting and eliminating asymptomatic malaria in different countries. The study is timely and relevant, as RACD is a key strategy in the malaria elimination program and considerable knowledge gaps exist regarding procedure optimization (10). The review provides useful information regarding risk assessment of acquiring malaria by  living with an index case of malaria within the household and within the neighborhood. Suggest analyzing implementation metrics of the reviewed articles with a view to procedural optimization as it would greatly enhance the usefulness of this review.

1.      Abstract, line 34, the sentence is not clear, “Systematic review of 54 studies and 7,13,and 29 respective studies ….”.need clarification regarding the “respective studies”.

2.      Abstract, line 35, Sentence need modification “The average risk of having malaria infection living….”

3.      Similarly under methods, the same sentence lines 120-123 “A systematic review……………..

4.      Table 1.  [18 ] Gunasekera et al , Sri Lanka, is classified as a low transmission region, However Sri Lanka was certified to have eliminated malaria as a Public Health Problem in 2016. Thus more appropriate would be no indigenous transmission.

5.      Can the authors explain the variables of Table 1, columns, 8 & 9 , RACD response per protocol and RACD response real time as content matter included in both are more or less similar. Are you testing adherence to protocol?

6.      Methods; Two previous published reviews on the same topic 19 & 20 were reported. Were they excluded from the current  review? Should include a statement to that effect.

7.      Pooled RACD data from different studies were utilized for  calculating, the average  risk of malaria infection among household members and neighbors. Pooling of quantitative data from regions with varying malaria endemicity may not truly reflect the risk of infection for a particular region as this will be affected by factors such as vector abundance, socioeconomic factors, malaria health policies in the regions etc and thus differ accordingly. Should include a statement to that effect under discussion.

8.      Analyzing implementation metrics ie screening radius, size of population / households screened, time of screening  in different settings  with their yield of positive cases (secondary cases) might provide useful data to optimize RACD in relevant in  settings. Explore the possibility of doing this as it might fill the knowledge gaps on RACD    

9.       Discussion seems somewhat  inadequate for achieving the stated objective of the study; discuss and highlight reactive case detection as a recommended strategy for detection and elimination of asymptomatic malaria in different countries.  A more country vise or regional focus to the discussion was anticipated by the stated objective. There is some degree of country / region vise discussion, but it is rather vague. Suggest enhancing this aspect in the discussion. The underlying factors for heterogeneity, variations of RACD implementation metrics according to malaria endemicity, urbanization, regional  malaria surveillance policies and other factors may improve the discussion and be more meaningful . 

10.   References, Reference 1 may be updated as World Malaria Report 2022 is available

Author Response

  1. Abstract, line 34, the sentence is not clear, “Systematic review of 54 studies and 7,13, and 29 respective studies….”.need clarification regarding the “respective studies”.

Response: Thanks very much for your comment. Clarification has been made in line 34 to 37

  1. Abstract, line 35, Sentence need modification “The average risk of having malaria infection living…

Response: Thanks very much for your comment. Modification has been made in line 37 to 39

  1. Similarly under methods, the same sentence lines 120-123 “A systematic review…

Response: Thanks very much for your comment. Clarification has been made in line 121 to 125

  1. Table 1.  [18 ] Gunasekera et al , Sri Lanka, is classified as a low transmissionregion, However Sri Lanka was certified to have eliminated malaria as a Public Health Problem in 2016. Thus more appropriate would be no indigenous transmission.

Response: Thanks very much. Correction has been effected in Reference 18 of line 129

  1. Can the authors explain the variables of Table 1, columns, 8 & 9 , RACD response per protocol and RACD response real time as content matter included in both are more or less similar. Are you testing adherence to protocol?

Response: Thanks very much. Adherence to protocol was not necessarily been tested. Column 8 is an extracted information of what the authors of the published papers sought to do prior to undertaking the study, while column 9 is what really happened practically or in reality during RACD. We have added REALITY to column 9 to clearly bring out the meaning.

  1. Methods; Two previous published reviews on the same topic 19 & 20 were reported. Were they excluded from the current review? Should include a statement to that effect.

Response: Thanks very much. The statement has been included in line 121

  1. Pooled RACD data from different studies were utilized for calculating, the average risk of malaria infection among household members and neighbors. Pooling of quantitative data from regions with varying malaria endemicity may not truly reflect the risk of infection for a particular region as this will be affected by factors such as vector abundance, socioeconomic factors, malaria health policies in the regions etc and thus differ accordingly. Should include a statement to that effect under discussion.

Response: Thanks very much. This statement has been included in line 245-250

  1. Analyzing implementation metrics ie screening radius, size of population / households screened, time of screening in different settings with their yield of positive cases (secondary cases) might provide useful data to optimize RACD in relevant in settings. Explore the possibility of doing this as it might fill the knowledge gaps on RACD.

Response: Thanks very much. This has been done in lines 269-272 and 301-315

  1. Discussion seems somewhat inadequate for achieving the stated objective of the study; discuss and highlight reactive case detection as a recommended strategy for detection and elimination of asymptomatic malaria in different countries. A more country vise or regional focus to the discussion was anticipated by the stated objective. There is some degree of country / region vise discussion, but it is rather vague. Suggest enhancing this aspect in the discussion. The underlying factors for heterogeneity, variations of RACD implementation metrics according to malaria endemicity, urbanization, regional malaria surveillance policies and other factors may improve the discussion and be more meaningful.

Response: Thanks for your comments. The discussion has been improved with the factors indicated above in lines 251-265, 269-282, 301-326 and 352-372

  1. References, Reference 1 may be updated as World Malaria Report 2022 is available

Response: Thanks for your comment. It has been updated as such and changes made to lines 51,52,55,56 and 57.